Convolutional neural networks for road surface classification on aerial imagery

Pesek Ondrej 1 ondrej.pesek@fsv.cvut.cz
Krisztian Lina 2
http://orcid.org/0000-0001-6869-3542 Landa Martin 1
Metz Markus 2
http://orcid.org/0000-0003-1916-1966 Neteler Markus 2
1 Department of Geomatics, Faculty of Civil Engineering, Czech Technical University in Prague , Prague , Czech Republic
2 Mundialis GmbH & Co. KG , Bonn, North Rhine-Westphalia , Germany
Schifanella Rossano
Electronic publication date: 2024 Dec 23
Publication date: 2024
Volume: 10
Electronic Location ID: e2571
Received 2024 Jun 17; Accepted 2024 Nov 11
Copyright: © 2024 Pesek et al.
Copyright year: 2024
Copyright holder: Pesek et al.
License: This is an open access article distributed under the terms of the Creative Commons Attribution License, which permits unrestricted use, distribution, reproduction and adaptation in any medium and for any purpose provided that it is properly attributed. For attribution, the original author(s), title, publication source (PeerJ Computer Science) and either DOI or URL of the article must be cited.
License URL: https://creativecommons.org/licenses/by/4.0/

Keywords: Remote sensing, Road surface, Convolutional neural network, CNN, Land cover detection

Funding: Grant Agency of the Czech Technical University SGS24/050/OHK1/1T/11 This work was supported by the Grant Agency of the Czech Technical University in Prague, grant No. SGS24/050/OHK1/1T/11. The funders had no role in study design, data collection and analysis, decision to publish, or preparation of the manuscript.

==============================
Any place the human species inhabits is inevitably modified by them. One of the first features that appear everywhere, in urban areas as well as in the countryside or deep forests, are roads. Further, roads and streets in general reflect their omnipresent and significant role in our lives through the flow of goods, people, and even culture and information. However, their contribution to the public is highly influenced by their surface. Yet, research on automated road surface classification from remotely sensed data is peculiarly scarce. This work investigates the capacities of chosen convolutional neural networks (fully convolutional network (FCN), U-Net, SegNet, DeepLabv3+) on this task. We find that convolutional neural network (CNN) are capable of distinguishing between compact (asphalt, concrete) and modular (paving stones, tiles) surfaces for both roads and sidewalks on aerial data of spatial resolution of 10 cm. U-Net proved its position as the best-performing model among the tested ones, reaching an overall accuracy of nearly 92%. Furthermore, we explore the influence of adding a near-infrared band to the basic red green blue (RGB) scenes and stress where it should be used and where avoided. Overfitting strategies such as dropout and data augmentation undergo the same examination and clearly show their pros and cons. Convolutional neural networks are also compared to single-pixel based random forests and show indisputable advantage of the context awareness in convolutional neural networks, U-Net reaching almost 25% higher accuracy than random forests. We conclude that convolutional neural networks and U-Net in particular should be considered as suitable approaches for automated semantic segmentation of road surfaces on aerial imagery, while common overfitting strategies should only be used under particular conditions.

Introduction

The flow of people, goods, and often even culture and information heavily depends on one feature of the urban landscape–roads. Roads play an important role in socio-economic growth by providing connection to resources, jobs, and markets (Verburg, Ellis & Letourneau, 2011) and due to their defining role in urban development patterns, they even predestine where and how will the human population live and grow (Barrington-Leigh & Millard-Ball, 2015). Moreover, road-based travels hold to be 80% of all transportation (Barrington-Leigh & Millard-Ball, 2017; Cuenot, Fulton & Staub, 2012).

It is then only natural that roads account for not only a significant aspect of humanity’s daily lives but also a notable area of Earth’s surface. It was quantified already in 2017 (Barrington-Leigh & Millard-Ball, 2017) that the global stock of roads totalled around 39.7×106 km, or around 5.6 m per person, about 134% of the International Road Federation (IRF) estimates. On top of that, the authors based the new estimate on data from the OpenStreetMap (OSM) project (https://www.openstreetmap.org/) (Haklay & Weber, 2008) which they considered to contain only about 83% of the global road network. Still, if a 1-km buffer was applied to these roads, only about 80% of the Earth’s terrestrial surface would remain roadless and would be fragmented into about 600,000 patches, more than half of which would be smaller than one square kilometre (Ibisch et al., 2016). Seeing that, it can be claimed that accurate and complete mapping of the global road infrastructure is essential to correctly define the status of the Earth’s surface and the roads’ impact on humans’ lives (Nelson, de Sherbinin & Pozzi, 2006).

The road’s location might be the most important information for researchers and commercial companies, but it is not the only attribute of interest. The fastest path computation uses average speed values assigned to each road link. The main inputs for the computation of these values are the road hierarchy level, slope, and its surface (Guth, Wursthorn & Keller, 2020). The road surface is directly used to compute the maximum road friction coefficient (Lee et al., 2021) which is together with the tire state crucial for vehicle braking systems (Yoo et al., 2022). Yet, the road surface information is beneficial also for other sides than just the transport and car industry. The road surface defines its drainage capacity and stiffness (Li et al., 2017) and affects animals’ behaviour towards roads (Assiss, Giacomini & Ribeiro, 2019). Asphalt roads have the potential of using TiO2 as a photocatalyst for NOx removal from vehicle emissions (Chen & Liu, 2010) while paving stones were found to be more effective in reducing urban heat islands than asphalt roads (Moretti et al., 2021). Compact surfaces such as asphalt and modular surfaces such as interlocking concrete blocks also profoundly differ in terms of maintenance, durability, sustainability, and stormwater management (Silva et al., 2023). Thus, the correct classification of distinct road surfaces holds a significant role in urban planning, transportation, and infrastructure maintenance.

The fact that the surface type is an important attribute of every road is underscored by its inclusion in multiple geospatial road datasets. Nevertheless—as far as the authors are aware—none of them satisfies either the spatial extent or the quality and completeness demands. The Global Roads Open Access Data Set might have three fields dedicated to the surface, but even the official documentation warns that only a few if any attributes are available for all road segments (Center for International Earth Science Information Network–CIESIN–Columbia University and Information Technology Outreach Services–ITOS–University of Georgia, 2013). OSM (Haklay & Weber, 2008)—another widely used source for geospatial data—also has an attribute for the road surface, but due to its crowd-sourcing nature, the data quality and the label usage are inconsistent and geographically uneven. Also, the cases of mixed surfaces and different surfaces on the road and the sidewalk are noticeably omitted. Therefore, the community could benefit from an automated method classifying larger geographic regions.

There are various methods for automated in-situ road surface classification. In-situ methods can be based on measuring acoustic noise signals from the tire cavity space (Masino et al., 2017) as well as on using convolutional neural networks (CNN) on a tire-pavement interaction noise from the wheel cover (Yoo et al., 2022) or on CNN-based camera data processing (Nolte, Kister & Maurer, 2018). However, the in-situ classification methods are too time and resources-consuming for an entire country mapping and very often serve only for the in-place vehicle behaviour instead. In lieu, as the camera-based approach has proved its place, remote sensing offers an option to map an entire area covered by aerial imagery in sufficient spectral/spatial detail in one automated step (Mohammadi, 2012).

An important information for the tasks mentioned above is whether the road surface is compact (asphalt, concrete; illustrated in Fig. 1A) or modular (sett, cobblestones; illustrated in Fig. 1B). In the area of remote sensing and computer vision, this differentiation can be hinted at not only by a specific pixel value but also by the overall texture and the objects’ shape and relationship with the surroundings. CNNs are a tool that was created explicitly to exploit such relationships and experienced growing success and popularity over the last several years (Razavian et al., 2014). CNNs have proved their strength in various remote sensing semantic segmentation applications such as urban vegetation classification (Torres et al., 2021), building segmentation (Alexakis & Armenakis, 2022), or cloud detection (Pešek, Segal-Rozenhaimer & Karnieli, 2022). Moreover, CNNs have shown promising results in road segmentation from remote sensing images (Cao, Zhang & Jiao, 2023; Ding et al., 2021) and even in road crack segmentation (Cao et al., 2023; Duan et al., 2024), a task useful for road surface classification. Despite that, their use in the remote sensing-based road surface classification itself is yet to be explored.

Figure 1 Examples of the compact and modular surface used in this chapter.

(Figure credit: https://commons.wikimedia.org/wiki/File:Mala_Hrastice_2020-06-16_Ulice_z_navsi_na_nadrazi_obr05.jpg; https://commons.wikimedia.org/wiki/File:Brusteinshalden.JPG).

The main objective of this article is to examine the capabilities of selected CNN architectures on the task of road surface classification. The selected architectures are fully convolutional network (FCN), U-Net, SegNet, and DeepLabv3+ and are chosen based on their utilisation frequency in the field of remote sensing. Positive findings can valuably contribute to the expanding discourse surrounding road surface material detection and to the remote sensing community itself. Moreover, this study examines the effect of using the near-infrared band and normalised difference vegetation index (NDVI) (Tucker et al., 1982) on the performance of a CNN.

Materials and Methods

The schematic processing workflow of this research is illustrated in Fig. 2. The following subsections will be dedicated to its components. “Study Area” presents the chosen study area. “Training Dataset” describes the training dataset and its process of creation. “Methods” briefly introduces the CNN architectures selected in this study. Having the training data and CNN architectures, the training itself could be run. The architectures ran multiple times in various arrangements of both the architecture and the training set, as described in “Data and CNN Settings”. Besides the performance over the training set, the models’ performance had to be validated over a validation set. The equations for the metrics used for this task are denoted in “Evaluation Metrics”.

Figure 2 The study workflow.

Study area

The study area consists of two locations in Bonn (Germany) and its vicinity. One is situated in Auerberg—an urban neighbourhood in the northern part of the city–, and the other one around Lessenich/Meßdorf—a considerably more rural area in the surroundings. The Auerberg scene visualised in Fig. 3C contains various types of roads, from primary roads through residential roads between houses and pedestrian streets in parks to special features such as parking lots and cycleway lanes. Very often, there are also roads surrounded by sidewalks, both being of a different surface type. On the other hand, the Meßdorf scene offers roads through vegetation and fields, as can be seen in Fig. 3D. Besides the variety of roads, the motivation for the study area was the authors’ place of residence. In cases of missing data or external data errors, it was possible to check the areas personally and fix the data issues.

Figure 3 Overview of the study area.

The data in subfigures (C) and (D) are coming from the North Rhine-Westphalia geoportal (available at https://www.geoportal.nrw) and are licensed under the DE-DL Zero 2.0 license (DL-DE, 2021).

Training dataset

The imagery data used in this study come from the North Rhine-Westphalia geoportal (https://www.geoportal.nrw). They are aerial orthophotos with 10-cm spatial resolution and four spectral bands (red, green, blue, near-infrared), retaken in a 2-year cycle.

As the potential label sources mentioned in “Introduction” were considered insufficient, a new training dataset had to be created. This was done semi-manually. The first draft of the data came from OSM, reclassifying the available palette of classes to one for compact surface (classes like asphalt, concrete, big concrete plates) and one for modular surface (classes like cobblestones, paving stones, sett). Because there were many gaps and sidewalks were completely missing, the classification of these had to be done manually. As anticipated, a lot of roads with the surface tag filled in OSM were found during the CNN trainings to be misclassified in OSM; these places were checked personally, fixes propagated back to OSM, and the models were retrained. Samples from the created dataset can be seen in Fig. 4.

Figure 4 Training dataset examples.

To examine the effect of the near-infrared band and NDVI (Tucker et al., 1982) on the road surface classification, three different input band sets were used for the training: Full-band scene (red, green, blue, and near-infrared);

Red green blue (RGB) scene;

RGB + NDVI scene.

Methods

In this study, two distinct methodologies were implemented. The first is a method centered on the individual values of pixels—random forests (Breiman, 2001)—while the second involves techniques that incorporate the context of neighbouring pixels—CNNs.

The random forest approach (Breiman, 2001) comprises a collective of decision trees that are independently trained and then aggregated into one decision algorithm. The application of the strong law of large numbers (Dekking et al., 2005) underpins its robustness against overfitting. Since its inception in 2001, random forests have been widely adopted across various fields and methodological frameworks (Liaw & Wiener, 2002).

CNNs became a solid part of remote sensing classification and segmentation methods and certain patterns and trends started to pop out. As can be seen in Fig. 5, the majority of CNN architectures used for semantic segmentation in the field of remote sensing is based on the so-called encoder-decoder (Ye & Sung, 2019) approach. The four most common encoder-decoder CNN architectures (Hoeser & Kuenzer, 2020; Hoeser, Bachofer & Kuenzer, 2020) were chosen to be utilised for the task of semantic segmentation in this study: FCN (Long, Shelhamer & Darrell, 2015), U-Net (Ronneberger, Fischer & Brox, 2015), SegNet (Badrinarayanan, Kendall & Cipolla, 2017), and DeepLabv3+ (Chen et al., 2018), described in “FCN”, “U-Net”, “SegNet”, and “DeepLabv3+”, respectively.

Figure 5 Most frequently used CNN models for per-pixel classification in remote sensing.

The four most frequent architectures utilising the encoder-decoder paradigm were utilised in this study: FCN, U-Net, SegNet, and DeepLab (Hoeser, Bachofer & Kuenzer, 2020).

The design of CNN architectures implemented in this study was kept as similar to their original proposals as possible, with only one greater enhancement. Every batch normalisation layer (Ioffe & Szegedy, 2017) is now followed by an optional (appearing in the model only if dropout rate defined as larger than 0) dropout layer (Hinton et al., 2012) in order to analyse its effect on overfitting, as utilised e.g., in Yang et al. (2019).

FCN

The FCN was introduced in 2014 (Long, Shelhamer & Darrell, 2015) as a tool for semantic segmentation of RGB photos. Although FCN was of a substantial size, it came into being as a pioneering convolutional network for semantic segmentation (He et al., 2020). The design of FCN can be seen in Fig. 6. It consists of a selected backbone architecture, fully-connected layers transformed into convolutional layers with a kernel size matching the feature map, and a 1×1 classifier convolutional layer. Depending on the FCN variant, classifications from different stage outputs are concatenated to the upsampled results of the deepest layer. FCN-32s utilises only the upsampled output of the last layer to match the original image size, FCN-16s adds concatenated classifications from the fourth stage, and FCN-8s also concatenates the classifications from the third stage. The original article used VGG16 (Simonyan & Zisserman, 2014) as the backbone architecture and so does this research.

Figure 6 A schematic representation of FCN-8s.

The input image (which is 352×352 in this study) is represented in orange, convolutions with activation functions in azure, max-pooling layers in red, sole convolutions with a kernel size matching the feature map in dark blue, upsampling layers in purple, concatenations in dark green, and finally the classification layer is in light green (Piramanayagam et al., 2018).

The variant specifically employed in this article is the VGG-16-backboned FCN-8s. It stands for the heaviest architecture in this research. The total parameter count for the FCN architecture used with the full-band dataset without dropout layers reached 1,260,241,575, out of which 1,260,213,671 were trainable.

U-Net

The schematic depiction of U-Net can be seen in Fig. 7. It is clear that its name comes from its symmetric, U-shaped encoder-decoder (Ye & Sung, 2019) based on the design of FCN (Long, Shelhamer & Darrell, 2015). The first addition to FCN was the incorporation of expanding decoder layers symmetric to those in the FCN’s sole encoder and connected by skip connections. That allows the context information to be propagated to layers representing higher-resolution images. This information is also enriched by entire feature maps transferred by skip connections from the corresponding encoder levels. The last change when compared to the FCN was a drop of the fully connected layers. This step made the models much lighter in terms of the parameter number. The architecture was originally proposed in 2015 to detect neuronal structures in electron microscopy images (Ronneberger, Fischer & Brox, 2015), but quickly got adopted by the computer vision and remote sensing community (Hoeser, Bachofer & Kuenzer, 2020).

Figure 7 U-Net architecture.

Source: (Hirose et al., 2022).

The total parameter count for the U-Net architecture used with the full-band dataset without dropout layers reached 31,056,003, out of which 31,044,227 were trainable; it consists of circa 494 GFLOPs.

SegNet

Briefly after the introduction of U-Net, another popular CNN was designed in 2015—SegNet (Badrinarayanan, Kendall & Cipolla, 2017). Figure 8 depicts the three differences from otherwise similar U-Net. First, its U-shaped encoder-decoder CNN structure contains two convolutional blocks at the deepest level instead of one. Second, one extra convolutional layer at its third and fourth levels. Third, skip layers transfer only pooling indices for their reuse instead of full feature map concatenation. The last difference actually saves a lot of resources and therefore compensates for the previous two ones.

Figure 8 SegNet architecture (Badrinarayanan, Kendall & Cipolla, 2017).

The total parameter count for the SegNet architecture used with the full-band dataset without dropout layers reached 62,502,595, out of which 62,481,603 were trainable; it consists of circa 438 GFLOPs.

DeepLabv3+

Although the symmetric encoder-decoder design is very popular, there are alternative approaches as well. Atrous (sparse, dilated) convolutions stay as one of the most common ones. They skip down- and upsampling layers by enlarging the receptive field by dilated kernels, thus keeping the original spatial resolution of the feature map. The atrous convolutions usually result in architectures with noticeably fewer parameters and therefore faster to be trained. The most popular family of this type of semantic segmentation performing CNNs according to Fig. 5 is the DeepLab family, consisting of the following generations: DeepLab (Chen et al., 2014), DeepLabv2 (Chen et al., 2017a), DeepLabv3 (Chen et al., 2017b), and finally the one chosen for this study, DeepLabv3+ (Chen et al., 2018).

Both the encoder and the decoder of the DeepLabv3+ architecture are modular. The encoder starts with a so-called backbone architecture, i.e., a classification layers-less CNN model. Three variants of ResNet (ResNet-50, ResNet-101, ResNet-151) (He et al., 2016) are utilised for this study, while the original article tested only ResNet-101 and Xception (Chollet, 2017). This module’s output then continues through an atrous spatial pyramid pooling (ASPP) and computational-complexity-reducing depthwise convolution called atrous separable convolution (Sifre, 2014). The decoder is connected to the encoder by a concatenation of the atrous separable convolution and the backbone-convoluted low-level outputs. The decoder then consists of a convolutional block and upsampling. The architecture is depicted in Fig. 9.

Figure 9 DeepLabv3+ architecture (Pedrayes et al., 2021).

The total parameter count and FLOPs for the DeepLabv3+ architecture used with the full-band dataset without dropout layers reached the following numbers, sorted by the backbone architecture: ResNet-50: 17,835,955, out of which 17,801,171 were trainable; circa 106 GFLOPs;

ResNet-101: 36,906,419, out of which 36,819,411 were trainable; circa 150 GFLOPs;

ResNet-152: 52,619,187, out of which 52,486,099 were trainable; circa 193 GFLOPs.

Data and CNN settings

The dataset consists of two 10,000×10,000 orthophoto tiles and corresponding labels. The classes were balanced in the tiles, with 11.7% pixels belonging to the compact surface class and 10.2% to the modular surface class. Both tiles were divided into patches of size 544×544 and only patches containing roads were selected, resulting in 187 patches of interest. A total of 60% of the patches were used for the model training and 40% for its validation to ensure sufficiently varied roads in both the training and validation datasets. Due to the time costs connected to data labelling, no data were used as extra test datasets. The model initialization method used in this study is the Glorot’s initialization technique (Glorot & Bengio, 2010). The maximum epoch was set to 2,500. However, this number was never reached as the early stopping was implemented to avoid overfitting; the patience was set to 100, meaning that every training was stopped if there was no improvement in loss for 100 consecutive epochs. Models were saved only when their validation loss value was lower than that any of the preceding epochs. Rectified linear units (ReLU) were utilised as the convolutional layers activation functions, the choice being motivated by their wide use in deep learning (Ramachandran, Zoph & Le, 2014). Every ReLU layer was followed by a batch normalisation one (Ioffe & Szegedy, 2017). The models’ optimiser was a first-order gradient-based optimiser of stochastic objective functions, Adam (Kingma & Ba, 2014).

Every CNN model ran four times on each of the datasets mentioned in “Training Dataset”; twice in different model settings, twice in different dataset settings. The model setting depended on the dropout rate—one training without dropout layers and one with a dropout rate of 0.5 (50%). The dataset setting depended on data augmentation—one run performed without data augmentation, one with simple data augmentation (using only rotations of 90, 180, and 270 degrees). Altogether, twelve trainings ran for each CNN model, as depicted in Table 1. The band aspects enetring the training are illustrated in Fig. 10. The effect of dropout and data augmentation on the results will be discussed in “Comparison among Methods” and “Comparison among Band Sets”, respectively.

Table 1 Model and dataset settings for each CNN architecture.

Dataset settings	
Model setting	Full-band	Augmented full-band	RGB	Augmented RGB	Full-band with NDVI	Augmented full-band with NDVI	
Model without dropout	True	True	True	True	True	True	
Model with dropout rate 50%	True	True	True	True	True	True	

Figure 10 Aspects entering the training datasets—Auerberg scene example.

The ideal configuration of the random forests algorithm for the task presented was determined through settings adjustments to be: Maximum depth of trees: 16; number of trees: 300; growing strategy: local.

Evaluation metrics

The trained models’ performance was evaluated using overall accuracy, F1 score, and Dice loss (used in the early stopping control).

Overall accuracy is defined as the fraction of correctly the predicted cases to the cases total amount (Metz, 1978), or, mathematically delineated:

(1) accuracy=TP+TNTP+TN+FP+FN,

where TP stays for the number of true positives, TN stays for the number of true negatives, FP stays for the number of false positives, and FN stays for the number of false negatives.

The performance of various methods is also measured using the F1 score, which is defined as the harmonic mean of precision and recall, with possible values ranging from 0 to 1 (Taha & Hanbury, 2015). An F1 score approaching 1 is the desired outcome. For the purpose of averaging the F1 score across all classes, this study employs macro averaging. The F1 score is mathematically expressed in the following manner:

(2) F1=2∗TP2∗TP+FP+FN,

where TP stays for the number of true positives, FP stays for the number of false negatives, and FN stays for the number of false negatives.

The metric Dice loss function is trying to minimise the level of association between compared classes, the so-called Sørensen–Dice coefficient. For two classes, it is defined as a ratio of twice the common area of two sets to the sum of their cardinalities (Dice, 1945).

(3) D=2∑iNgipi∑iNgi2+∑iNpi2,

where D is the Sørensen–Dice coefficient, N is the number of training/validation samples, gi is the count of ground-truth pixels for sample i, and pi is the count of predicted pixels for sample i.

Defined in 1945 to quantify the degree of association between two distinct species in nature, this concept has found applicability in expressing the association among various sets of data. Hence, it entered the realm of artificial neural networks (ANNs) as early as the 1990s (Zijdenbos et al., 1994). Unlike certain loss functions such as binary cross-entropy that operate with the total count of pixels per class, potentially resulting in bias towards classes with fewer pixels, Dice loss mitigates the impact of imbalanced classes by computing the loss function relatively for each class. This rationale results in its wide use in multi-class segmentation problems.

Results and discussion

Results for different models and configurations are presented and discussed in this section. Overall accuracy values for all CNNs and data settings used in this study are reported in “Comparison among Methods”, where the performance of various CNN architectures and the impact of dropout are discussed, too. “Comparison among Band Sets” details a comparison of the performance of the best architecture across different band sets and the effect of the simple data augmentation.

Comparison among methods

This section consists of the comparison of results from various CNN architectures, specifically FCN, U-Net, SegNet, and DeepLabv3+, and random forests algorithm. Overall accuracy values for all settings are reported in Table 2, corresponding Dice loss values in Table 3, corresponding F1 score in Table 4, and corresponding mean over union values in Table 5. These values were calculated using the independent validation dataset, as detailed in “Data and CNN Settings”. The patches used for visualisations throughout this section are chosen from those employed for the overall accuracy computation.

Table 2 Overall accuracy values (in per cent) over the validation dataset for different architectures and settings.

For rows: dN% stands for dropout, N specifies the dropout ratio, and rnX specifies the ResNet version used as a backbone model for DeepLabv3+. For columns: fb stands for the dataset utilising the full-band images, rgb stands for the dataset utilising only the red, green, and blue bands, fb_ndvi represents the full-band dataset enhanced by the NDVI, and _a represents the augmented version of the dataset. The highest overall accuracy value is typed in bold. The table layout is the same as used in Pešek, Segal-Rozenhaimer & Karnieli (2022).

Architecture	fb	fb_a	rgb	rgb_a	fb_ndvi	fb_ndvi_a	
FCN_d00%	89.5	89.7	89.4	87.0	89.2	90.2	
FCN_d50%	71.4	83.2	80.9	67.8	74.2	73.0	
U-Net_d00%	91.1	89.2	87.6	90.3	90.9	89.6	
U-Net_d50%	77.6	67.7	69.1	71.2	79.8	66.8	
SegNet_d00%	90.3	88.1	86.8	87.8	89.6	88.8	
SegNet_d50%	66.7	83.7	80.1	75.4	79.2	84.6	
DeepLabV3+_rn50_d00%	81.2	79.7	81.3	86.9	75.4	84.7	
DeepLabV3+_rn50_d50%	67.2	69.3	70.9	81.9	66.5	63.1	
DeepLabV3+_rn101_d00%	75.9	82.6	80.0	84.9	83.9	86.2	
DeepLabV3+_rn101_d50%	69.0	65.7	83.2	81.7	62.6	48.1	
DeepLabV3+_rn152_d00%	77.4	85.9	84.4	83.2	80.8	74.0	
DeepLabV3+_rn152_d50%	75.3	69.8	79.3	82.1	63.8	70.4	
random_forests	59.4	59.4	65.5	66.5	60.5	60.5	

Table 3 Dice loss values over the validation dataset for different architectures and settings.

For rows: dN% stands for dropout, N specifies the dropout ratio, and rnX specifies the ResNet version used as a backbone model for DeepLabv3+. For columns: fb stands for the dataset utilising the full-band images, rgb stands for the dataset utilising only the red, green, and blue bands, fb_ndvi represents the full-band dataset enhanced by the NDVI, and _a represents the augmented version of the dataset. The lowest loss function value is typed in bold. The table layout is the same as used in Pešek, Segal-Rozenhaimer & Karnieli (2022).

Architecture	fb	fb_a	rgb	rgb_a	fb_ndvi	fb_ndvi_a	
FCN_d00%	0.470	0.444	0.469	0.472	0.482	0.436	
FCN_d50%	0.631	0.662	0.635	0.667	0.643	0.654	
U-Net_d00%	0.436	0.440	0.480	0.444	0.442	0.426	
U-Net_d50%	0.580	0.608	0.628	0.633	0.600	0.639	
SegNet_d00%	0.470	0.465	0.498	0.492	0.483	0.491	
SegNet_d50%	0.637	0.627	0.617	0.626	0.622	0.601	
DeepLabV3+_rn50_d00%	0.549	0.547	0.543	0.572	0.562	0.527	
DeepLabV3+_rn50_d50%	0.638	0.623	0.628	0.666	0.672	0.688	
DeepLabV3+_rn101_d00%	0.564	0.518	0.525	0.525	0.548	0.524	
DeepLabV3+_rn101_d50%	0.614	0.609	0.670	0.672	0.679	0.732	
DeepLabV3+_rn152_d00%	0.559	0.530	0.522	0.523	0.533	0.541	
DeepLabV3+_rn152_d50%	0.573	0.594	0.627	0.668	0.665	0.630	

Table 4 F1 score values over the validation dataset for different architectures and settings.

For rows: dN% stands for dropout, N specifies the dropout ratio, and rnX specifies the ResNet version used as a backbone model for DeepLabv3+. For columns: fb stands for the dataset utilising the full-band images, rgb stands for the dataset utilising only the red, green, and blue bands, fb_ndvi represents the full-band dataset enhanced by the NDVI, and _a represents the augmented version of the dataset. The highest F1 score is typed in bold. The table layout is the same as used in Pešek, Segal-Rozenhaimer & Karnieli (2022).

Architecture	fb	fb_a	rgb	rgb_a	fb_ndvi	fb_ndvi_a	
FCN_d00%	0.536	0.570	0.560	0.479	0.570	0.637	
FCN_d50%	0.452	0.195	0.381	0.414	0.435	0.343	
U-Net_d00%	0.686	0.609	0.708	0.464	0.645	0.499	
U-Net_d50%	0.335	0.396	0.273	0.391	0.266	0.312	
SegNet_d00%	0.507	0.471	0.489	0.460	0.492	0.425	
SegNet_d50%	0.140	0.341	0.040	0.182	0.294	0.307	
DeepLabV3+_rn50_d00%	0.478	0.442	0.460	0.431	0.474	0.405	
DeepLabV3+_rn50_d50%	0.399	0.366	0.397	0.359	0.390	0.320	
DeepLabV3+_rn101_d00%	0.493	0.464	0.480	0.445	0.486	0.425	
DeepLabV3+_rn101_d50%	0.417	0.378	0.409	0.379	0.398	0.337	
DeepLabV3+_rn152_d00%	0.426	0.467	0.485	0.454	0.488	0.433	
DeepLabV3+_rn152_d50%	0.422	0.381	0.410	0.384	0.408	0.344	
random_forests	0.323	0.323	0.143	0.143	0.387	0.387	

Table 5 Mean intersection over union values over the validation dataset for different architectures and settings.

For rows: dN% stands for dropout, N specifies the dropout ratio, and rnX specifies the ResNet version used as a backbone model for DeepLabv3+. For columns: fb stands for the dataset utilising the full-band images, rgb stands for the dataset utilising only the red, green, and blue bands, fb_ndvi represents the full-band dataset enhanced by the NDVI, and _a represents the augmented version of the dataset. The highest value is typed in bold. The table layout is the same as used in Pešek, Segal-Rozenhaimer & Karnieli (2022).

Architecture	fb	fb_a	rgb	rgb_a	fb_ndvi	fb_ndvi_a	
FCN_d00%	0.689	0.830	0.443	0.717	0.643	0.773	
FCN_d50%	0.339	0.334	0.334	0.334	0.302	0.323	
U-Net_d00%	0.868	0.869	0.858	0.839	0.876	0.813	
U-Net_d50%	0.789	0.665	0.381	0.750	0.326	0.634	
SegNet_d00%	0.426	0.784	0.822	0.758	0.645	0.730	
SegNet_d50%	0.758	0.725	0.565	0.341	0.442	0.806	
DeepLabV3+_rn50_d00%	0.662	0.793	0.429	0.794	0.537	0.768	
DeepLabV3+_rn50_d50%	0.342	0.626	0.331	0.784	0.318	0.306	
DeepLabV3+_rn101_d00%	0.525	0.805	0.766	0.796	0.426	0.815	
DeepLabV3+_rn101_d50%	0.331	0.664	0.710	0.757	0.318	0.245	
DeepLabV3+_rn152_d00%	0.356	0.807	0.582	0.749	0.751	0.776	
DeepLabV3+_rn152_d50%	0.421	0.633	0.712	0.725	0.409	0.659	
random_forests	0.302	0.302	0.335	0.335	0.307	0.307	

The best result was reached by U-Net, as can be seen in Tables 2–5. The reported values show U-Net’s superiority to other models in terms of all the reported metrics. Visual checks illustrated in Fig. 11 and “A More Architecture Performance Visualisations” confirm the position of U-Net as the most suitable architecture for the task of road surface semantic segmentation among those tested in this article.

Figure 11 Semantic segmentation on a chosen validation data patch from various architectural settings, the Auerberg scene.

The parameter count of U-Net is lower than the one of FCN, SegNet, and two of the tested DeepLabv3+ variants. Its overall rule says that the design of a CNN model is more important than boosting it with the highest parameter count possible. But the depth still plays an important role in catching complex features if there is enough training data, as can be seen from the fact that the best-performing version of DeepLabv3+ is the deepest one. DeepLabv3+ with the ResNet-152 backbone outperforms the ResNet-50 based one in eight cases out of 12 in terms of overall accuracy and in nine cases out of 12 in terms of Dice loss; it also outperforms the ResNet-101 based DeepLabv3+ in eight cases out of 12 in terms of overall accuracy and 10 cases out of 12 in terms of Dice loss.

Comparisons with single pixel based random forests (Breiman, 2001) algorithm underpin the superiority of CNNs for the task of road surface semantic segmentation. There is only one case (DeeplabV3+ with ResNet 101 backbone and dropout 50%) when random forests reach higher overall accuracy than any of the CNNs and although random forests can reach higher F1 in some occasions, these are still marginal cases. While U-Net reaches at its best overall accuracy 91.1%, random forests reach 65.5%; similarly, while the highest F1 score for U-Net is 0.708, it is only 0.387 for random forests. An extreme case of the reason for random forests’ failure is clearly visible in Fig. 11. The lack of the neighbourhood information leads to the modular surface (where the relationship between neighbouring pixels is crucial) being very underdetected in favour of the compact surface. Another effect is poor distinction of road and non-road features with similar colours on top, such as flat concrete roofs.

In 97% of cases (35 out of 36), the dropout layers utilisation led to lower overall accuracy and in 100% of cases to higher loss value. Although this behaviour has not been expected, it is not a surprising finding. The overfitting threat decreases with the dataset growth, and such behaviour was already seen in other studies (Pešek, Segal-Rozenhaimer & Karnieli, 2022). Moreover, the distinction between the two classes of interest lies in the smallest details and turning the corresponding neurons off could have a crucial negative effect on the model’s performance. This finding underlines the best practice saying that researchers working with CNNs should first see if the model tends to overfit and only after that decide if dropout should be applied or not.

The visualisations in Fig. 11 not only prove the position of U-Net but also illustrate the regressing effects of dropout and show the common areas of misclassifications. The most common misclassification happens in areas covered with shadows which are either not classified at all or misclassified (in favour of the compact class for compact shadows, in favour of modular otherwise). SegNet can be seen to oversegment the images and FCN tends to smoothen the labels, resulting in very bad differentiation between roads and sidewalks. DeepLabv3+ variants usually suffer from flat roofs as being detected as roads.

Comparison among band sets

The common approach is using all bands available as they could bear useful information. However, the information carried in the band is not necessarily important for every task. In some cases, the information can confuse the model as it tries to conclude some decisions from non-relevant values, and in some cases, it could help the model in certain aspects and make it worse in others. It was found that the latter is the case for the road surface classification.

It is visible in Fig. 12 and particularly in Fig. 12D that the omission of the infrared band can lead to misclassification in partially shaded areas due to their big colour changes. On the other hand, the infrared band bears rich information on the recognition of vegetated and non-vegetated areas; its inclusion leads to a large bias towards any non-vegetated area to be interpreted as a road, resulting in incorrect classification of line-shaped parts of roofs, garages, and front yards. This double-sided nature of the infrared band is illustrated also in Tables 2 and 3 by the fact that the band inclusion led only in 38% of the 24 cases to higher overall accuracy, but in 58% to lower Dice loss value. The choice of using the infrared band should therefore be based on the chosen pre- or postprocessing (shadows filtering, masking out non-road areas, etc.,) and input data/areas of interest. The addition of NDVI to the full-band dataset actually led in 71% (17 cases out of 24) to higher loss value. As can be seen in Fig. 12, this is probably due to NDVI magnifying the vegetation vs non-vegetation road-decision paradigm.

Figure 12 Semantic segmentation on a chosen validation data patch from various inputs using U-Net–Dropout 0%.

Another finding is that using the simple right angle rotations as a means of data augmentation led to higher overall accuracy only in 18 out of 36 cases (and in 53% cases to higher loss value). This finding was surprising at first as the augmentation multiplied the training dataset size by four but visual inspection revealed that its effect was again double-sided. Although it helped with certain aspects such as border refining and rare objects and thin sidewalks classification, the models started to exclude many roads covered by shadows. The shaded roads omission is believed to be caused by the fact that the shadows are not rotation-independent in the area of interest (western Germany, northern hemisphere) but hold a northern-bound direction instead; areas near the equator should not suffer from this handicap. By training the model on data where this direction is unnaturally modified, it ceases to have the option to learn this feature that helps to correctly recognize the shadows. This behaviour can be seen in the lower right quarter of Fig. 12. A research investigating other types of data augmentation methods would be a valuable addition to the discussion.

Conclusion

Although the road surface classes are obviously important features for many real-world applications, a tool for its automated detection in remotely sensed data is perceptibly missing. This study explores this area with open-source tools and on open aerial data. It does so by using the most popular CNN architectures and benchmarking them on a manually created dataset. It proves that CNNs are capable of distinguishing between modular and compact road surface classes. It was found that the best-performing from the chosen architectures was U-Net, reaching almost 92% overall accuracy and a Dice loss value of around 0.4. Although the models perform well on clearly visible roads, their performance rapidly decreases with rarer features that do not have the typical road shape (e.g., squares) and areas covered by scattered shadows and tree crowns. The superior performance of CNNs for road surface semantic segmentation was made evident by comparisons with results obtained using the random forests algorithm.

Albeit the dataset is big enough to lower the threat of overfitting, two typical methods intended for overfitting amelioration were examined in dedicated experiments. The first such approach was the use of dropout layers, the second was simple right-angle-rotations data augmentation. The use of dropout layers actually led to lower overall accuracy. This is believed to be caused by the fact that the difference between modular and compact classes happens on the level of individual pixels instead of higher-level objects and turning off individual neurons in the model could be crucial for this distinction. The latter—the simple data augmentation method—has a very ambivalent effect and its use should be based on pre- or postprocessing applied to the data. While it helped to refine certain (thin, rare) features, it made the model lose the direction-dependent knowledge on some features and made the effect of shadows even more critical.

In addition, an experiment with a band set reduction and NDVI inclusion was conducted. A double-sided effect similar to that of the rotation data augmentation was caused by omitting the infrared band. While the inclusion of the infrared band helps to ameliorate the shadows’ negative impact, it also overly accentuates the vegetated—nonvegetated distinction, leading to line-shaped parts of roofs being occasionally misclassified as roads. The inclusion of NDVI magnified the effect of the infrared band.

Although CNNs in this article—and U-Net in particular—spearhead the task of automated road surface classification on remote sensing data, their use does not come without their typical downsides. For a long time, their main downside was their black box character. Albeit still being one of their weaknesses, its position has been shaken by the advent of explainable artificial intelligence (XAI) (Gunning & Aha, 2019). XAI helps to interpret the inner relationships of neural networks and a deep research on XAI and road surface classification could shine new, valuable light on the remote sensing-based road surface classification. The other issue that needs to be mentioned here is the computational needs of CNNs. The models ran on nVidia Tesla V100-32GB GPU and although it was around 80 times faster than running it on 20 CPU cores, the training of the heaviest model (FCN) on the augmented dataset took 9 days, making it a very time-consuming process for benchmarks and experiments. Beyond the general limitations of the proposed models, a certain limitation of the presented research needs to be mentioned. The presented models performed well in the area of interest but that does not mean that they would perform with the same accuracy in different areas. A research extending the training dataset to contain other areas and other types of roads would be a valuable addition.

Where possible, this article adheres to open science principles (Vicente-Saez & Martinez-Fuentes, 2018). The code (Pešek, 2024a) used in this article is available under the MIT license (MIT, 1987), while the data (Pešek, 2024b) are accessible under the Creative Commons Attributions 4.0 International license (CC-BY-4, 2013). As the band and index subsets could be obtained from the full-band one, only the full-band dataset is uploaded.

Appendices

A more architecture performance visualisations (Fig. 13)

Figure 13 Semantic segmentation on a chosen validation data patch from various architectural settings, the Meßdorf scene.

Additional Information and Declarations

Competing Interests

Author Contributions

Data Availability

The authors declare that they have no competing interests. Lina Krisztian, Markus Metz, and Markus Neteler are employed by mundialis GmbH & Co. KG.

Ondrej Pesek conceived and designed the experiments, performed the experiments, analyzed the data, performed the computation work, prepared figures and/or tables, authored or reviewed drafts of the article, and approved the final draft.

Lina Krisztian conceived and designed the experiments, authored or reviewed drafts of the article, and approved the final draft.

Martin Landa conceived and designed the experiments, authored or reviewed drafts of the article, and approved the final draft.

Markus Metz conceived and designed the experiments, authored or reviewed drafts of the article, and approved the final draft.

Markus Neteler conceived and designed the experiments, authored or reviewed drafts of the article, and approved the final draft.

The following information was supplied regarding data availability:

The training and testing dataset is available at Zenodo: Pesek, O. (2024). Road surface aerial photo training dataset (1.0) [Data set]. Zenodo. https://doi.org/10.5281/zenodo.10602515.

The code used in the experiments is available at GitHub and Zenodo: https://github.com/pesekon2/road-surface-detection-aerial.

Pesek, O. (2024). Convolutional Neural Networks for Road Surface Classification on Aerial Imagery (1.0). Zenodo. https://doi.org/10.5281/zenodo.14271123.

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
