# Peer review of "Convolutional neural networks for road surface classification on aerial imagery"

_PeerJ Computer Science, doi:10.7717/peerj-cs.2571_

## Round 0.1 · original submission · Major Revisions

The reviewers suggested several open points that need to be fully addressed for the manuscript to be considered for publication. Please address each comment and clearly explain in the response letter how you have addressed them and the modifications you made in the manuscript.

In particular, focus on (not exhaustive list):

1. Provide a more comprehensive framing and context in the Introduction and Background section, especially in relation to state-of-the-art methods in the area such as road crack segmentation.
2. Emphasize and underline the methodological contributions of the paper to make it clearer what this paper adds to the existing approaches.
3. Add or better explain several technical details in the experimental part as per the reviewers' comments.
4. Ensure the manuscript includes a proper discussion of the approach's limitations, impact, and future directions.
5. Improve the presentation quality through careful proofreading.

Reviewer 1 ·

Basic reporting

The introduction and background part is not making sence. since your work is focus on the segmenting of roads from remote images, cite more work on this, rather than accentuating the road is important.

In Figure 7, the input image is not a road surface image. please fix it.

Table 1 please give each column a name under the Dataset settings.

Experimental design

This paper is more like an application of existing CNNs on the segmentation of roads. there is no original research or methods proposed by the authors.

Validity of the findings

I suggest the authors can show the results for the impact of drop out rate and the impact of different CNN model, seperately, rather than display them together. it will make the results more clear.
Also, please visulize more results rather than only use one sample image (the authors can resize it to a small size or appendix it).

Additional comments

no comments.

Cite this review as

Reviewer 2 ·

Basic reporting

The paper presents a compelling exploration of automated road surface classification using convolutional neural networks (CNNs) on remotely sensed aerial data. It addresses a critical gap in the research, as the classification of road surfaces is an area with significant practical applications yet remains underexplored. By evaluating the performance of various CNN architectures—specifically FCN, U-Net, SegNet, and DeepLabv3+—the study contributes valuable insights into the potential of these models to distinguish between compact and modular surfaces on roads and sidewalks.

The study's methodology is well-structured, with a clear focus on comparing the performance of different CNN models. The contribution is quite OK. However, a small point to correct is that the result for semantic segmentation on a chosen validation data patch should be visually added and discussed more deeply and obviously. The literature review on pixel-level thin crack detection on road surfaces should be added, particularly using CNN and AutoML. In this way, I recommend accepting the paper with minor revisions to address the identified areas for improvement.

Experimental design

The paper presents a compelling exploration of automated road surface classification using convolutional neural networks (CNNs) on remotely sensed aerial data. It addresses a critical gap in the research, as the classification of road surfaces is an area with significant practical applications yet remains underexplored. By evaluating the performance of various CNN architectures—specifically FCN, U-Net, SegNet, and DeepLabv3+—the study contributes valuable insights into the potential of these models to distinguish between compact and modular surfaces on roads and sidewalks.

The study's methodology is well-structured, with a clear focus on comparing the performance of different CNN models. The contribution is quite OK. However, a small point to correct is that the result for semantic segmentation on a chosen validation data patch should be visually added and discussed more deeply and obviously. The literature review on pixel-level thin crack detection on road surfaces should be added, particularly using CNN and AutoML. In this way, I recommend accepting the paper with minor revisions to address the identified areas for improvement.

Validity of the findings

The paper presents a compelling exploration of automated road surface classification using convolutional neural networks (CNNs) on remotely sensed aerial data. It addresses a critical gap in the research, as the classification of road surfaces is an area with significant practical applications yet remains underexplored. By evaluating the performance of various CNN architectures—specifically FCN, U-Net, SegNet, and DeepLabv3+—the study contributes valuable insights into the potential of these models to distinguish between compact and modular surfaces on roads and sidewalks.

The study's methodology is well-structured, with a clear focus on comparing the performance of different CNN models. The contribution is quite OK. However, a small point to correct is that the result for semantic segmentation on a chosen validation data patch should be visually added and discussed more deeply and obviously. The literature review on pixel-level thin crack detection on road surfaces should be added, particularly using CNN and AutoML. In this way, I recommend accepting the paper with minor revisions to address the identified areas for improvement.

Cite this review as

Reviewer 3 ·

Basic reporting

1. The introduction is not good enough and the authors did not have a thorough literature review. Please add more newer references about crack segmentation.
2. Numerous works have been done for crack segmentation. The authors used existing models for crack segmentation. What are the innovations and contributions of this work? What are the differences between this work and previous work?
3. This work is about crack segmentation. However, the authors mentioned crack classification in the title. Please clarify it.

Experimental design

4. Please add image samples about 1) Full-band scene (red, green, blue, and near-infrared); 2) RGB scene; and 3) RGB + NDVI scene.
5. Please add Mean Intersection over Union (mIoU) in the evaluation metrics. This is the most popular evaluation metric for segmentation.
6. Please add the model and computational complexity of models, including the number of parameters, floating point operations (FLOPs), and frames per second (FPS).
7. 60% is the training and 40% is the validation on Page 8. Did the authors use the test dataset? Please clarify it.
8. What is the meaning of models with or without dropout rate? Did the authors add a dropout layer in models?

Validity of the findings

9. What is the limitation of the proposed models? Please add the limitations and future works in the conclusion.
10. Many typos and grammatical mistakes can be found in the manuscript. Thorough proofreading should be performed before resubmission.

Cite this review as

---

## Round 0.2 · accepted · Accept

The reviewers were satisfied with the explanations and modifications in the revised manuscript. We suggest performing additional overall proofreading, while in production.

Reviewer 2 ·

Basic reporting

The manuscript has been improved, so I recommend accepting it.

Experimental design

The manuscript has been improved, so I recommend accepting it.

Validity of the findings

The manuscript has been improved, so I recommend accepting it.

Cite this review as

Reviewer 3 ·

Basic reporting

Good to me

Experimental design

Good to me

Validity of the findings

Good to me

Additional comments

Good to me

Cite this review as